# Anti-Inflammatory Effect of Allicin Associated with Fibrosis in Pulmonary Arterial Hypertension

**DOI:** 10.3390/ijms22168600

**Published:** 2021-08-10

**Authors:** José L. Sánchez-Gloria, Constanza Estefanía Martínez-Olivares, Pedro Rojas-Morales, Rogelio Hernández-Pando, Roxana Carbó, Ivan Rubio-Gayosso, Abraham S. Arellano-Buendía, Karla M. Rada, Fausto Sánchez-Muñoz, Horacio Osorio-Alonso

**Affiliations:** 1Sección de Estudios de Posgrado, Escuela Superior de Medicina, Instituto Politécnico Nacional, Mexico City 11340, Mexico; luis_san29@hotmail.com (J.L.S.-G.); aiorubio@gmail.com (I.R.-G.); 2Departamento de Inmunología, Instituto Nacional de Cardiología Ignacio Chávez, Mexico City 14080, Mexico; michelrp36@gmail.com; 3Sección de Patología Experimental, Departamento de Patología, Instituto Nacional de Ciencias Médicas y Nutrición Salvador Zubiran, Mexico City 14080, Mexico; constanzamtz@hotmail.com (C.E.M.-O.); rogelio.hernandezp@incmnsz.mx (R.H.-P.); 4Departamento de Fisiopatología Cardio-Renal, Instituto Nacional de Cardiología Ignacio Chávez, Mexico City 14080, Mexico; pedrorojasm@outlook.com (P.R.-M.); abraham.arellano@cardiologia.org.mx (A.S.A.-B.); 5Departamento de Biomedicina Cardiovascular, Instituto Nacional de Cardiología Ignacio Chávez, Mexico City 14080, Mexico; roxana.carbo@cardiologia.org.mx

**Keywords:** pulmonary arterial hypertension, inflammation, fibrosis, allicin

## Abstract

Pulmonary arterial hypertension (PAH) is characterized by pulmonary vascular remodeling. Recent evidence supports that inflammation plays a key role in triggering and maintaining pulmonary vascular remodeling. Recent studies have shown that garlic extract has protective effects in PAH, but the precise role of allicin, a compound derived from garlic, is unknown. Thus, we used allicin to evaluate its effects on inflammation and fibrosis in PAH. Male Wistar rats were divided into three groups: control (CON), monocrotaline (60 mg/kg) (MCT), and MCT plus allicin (16 mg/kg/oral gavage) (MCT + A). Right ventricle (RV) hypertrophy and pulmonary arterial medial wall thickness were determined. IL-1β, IL-6, TNF-α, NFκB p65, Iκβ, TGF-β, and α-SMA were determined by Western blot analysis. In addition, TNF-α and TGF-β were determined by immunohistochemistry, and miR-21-5p and mRNA expressions of *Cd68*, *Bmpr2*, and *Smad5* were determined by RT-qPCR. Results: Allicin prevented increases in vessel wall thickness due to TNF-α, IL-6, IL-1β, and Cd68 in the lung. In addition, TGF-β, α-SMA, and fibrosis were lower in the MCT + A group compared with the MCT group. In the RV, allicin prevented increases in TNF-α, IL-6, and TGF-β. These observations suggest that, through the modulation of proinflammatory and profibrotic markers in the lung and heart, allicin delays the progression of PAH.

## 1. Introduction

Pulmonary arterial hypertension (PAH) is a life-threatening disease characterized by the progressive loss and obstructive remodeling of the pulmonary vascular bed. PAH leads to a progressive elevation in pulmonary arterial pressure (PAP) and pulmonary vascular resistance (PVR), resulting in functional decline and right heart failure [1]. The poor clinical outcome in patients with PAH is determined by the adaptation of right ventricle (RV) function to the increased afterload mediated by the increased contractility with preserved dimensions and stroke volume [2]. Currently, the pathogenesis of PAH remains unclear and involves numerous factors, including endothelial dysfunction, oxidative stress, and the exaggerated infiltration of inflammatory cells, as well as alterations in signaling pathways to maintain cellular identity and functionality as the morphogenetic protein receptor (BMPR2) [3,4].

In clinical practice, PAH treatments are limited to the production of vasoactive substances such as nitric oxide (NO) and prostacyclin [5]. However, numerous studies have demonstrated that inflammation is associated with the development of experimental and human PAH [6]. The inflammatory process leads to endothelial cell injury and stimulates pulmonary arterial smooth muscle cell (PASMC) proliferation, playing a key role in triggering and maintaining pulmonary vascular remodeling [7,8]. In fact, in experimental models, increased levels of interleukin-1β (IL-1β), IL-6, tumor necrosis factor-α (TNF-α), and nuclear factor-κB (NFκB) were observed [9]. On the other hand, TNF-α inhibition showed a favorable effect on hemodynamics and pulmonary vascular remodeling in experimental PAH [10]. Thus, this evidence supports the role of inflammation in the pathogenesis and progression of PAH. On the other hand, an increased transforming growth factor beta (TGF-β) protein expression can result in anti-apoptotic responses in pulmonary artery endothelial cells (PAEC) and PASMC [11] and can be a hallmark of endothelial cell (EC) dysfunction [12]. In addition, TGF-β can increase the presence of inflammatory markers [13]. Evidence suggests that cytokines and growth factors play a crucial role in hypertrophy and cardiac fibrosis [14]. Understanding the mechanisms promoting this pathologic process is essential to develop new therapeutic options to reduce RV failure and mortality in PAH patients.

Several studies have reported that medicinal plants, nutraceuticals, and phytochemicals exert significant benefits for PAH [15]. These natural compounds have anti-inflammatory, antiproliferative, and anti-vascular remodeling properties [16].

Garlic (*Allium sativum L.*) and its derived products have been widely used for culinary and medicinal purposes in many cultures and civilizations [17]. Garlic extract has protective effects in PAH, but its molecular mechanism is unknown [18]. The compounds present in garlic include protein, carbohydrates, vitamins, and minerals. Garlic is also especially rich in sulfur compounds, such as alliin, ajoenes, sulfides, and disulfides [19].

Allicin is a natural compound produced from the stable precursor S-allyl cysteine-sulfoxide (alliin) by the action of the enzyme alliinase when garlic cloves are crushed or macerated [20]. This compound has shown various beneficial effects, such as antioxidant and anti-inflammatory effects in cardiovascular diseases [21,22].

Allicin exerts its anti-inflammatory effects through several mechanisms. However, in PAH, these mechanisms remain poorly studied or unknown. Thus, the present study aimed to assess if allicin may exert beneficial effects in the progression of experimental PAH. Our results showed that allicin administration had a protective effect in the MCT model through the prevention of RV hypertrophy and increased pulmonary arterial medial wall thickness. In addition, through the modulation of TNF-α, IL-6, IL-1β, TGF-β, and α-SMA, allicin prevented inflammation and fibrosis in lung tissue. The increases in TNF-α, IL-6, and TGF-β in RV tissue were prevented by the allicin treatment. Therefore, these results suggest that allicin protects the RV, which is one of the heart chambers closely related to the severity and progression of PAH.

## 2. Results

### 2.1. Effect of Allicin on Right Ventricle (RV) Hypertrophy and Lung Morphology (Pulmonary Arteries Wall)

RV hypertrophy is characteristic of PAH in humans. Therefore, we assessed RV hypertrophy using the Fulton index (RV/LV + S) in the experimental model as an indicator of MCT-induced PAH. First, we showed that MCT induced RV hypertrophy compared with the control group (*p* < 0.05). We found that RV hypertrophy in the allicin-treated group was lower compared with the MCT group (Figure 1A). Second, we determined the effect of allicin on lung vascular remodeling. We showed that the pulmonary vessel wall was thickened and that the lumen was narrower in the MCT group compared with the control group (Figure 1B,C) (*p* < 0.05). Allicin treatment prevented an increase in vessel wall thickness when compared with the untreated group (Figure 1B). Altogether, these data demonstrate that the PAH model was successfully developed.

### 2.2. Anti-Inflammatory Effect of Allicin

In order to identify the anti-inflammatory effect of allicin on MCT-induced PAH, the expressions of inflammatory markers such as Cd68, TNF-α, IL-1β, and IL-6 were determined. First, we found that MCT significantly upregulated Cd68, TNF-α, IL-6, and IL-1β compared with the control group (*p* < 0.05) (Figure 2A,B and Figure 3A,B). The expressions of these inflammatory markers were lower in the allicin-treated group compared with the MCT group (*p* < 0.05) (Figure 2A,B and Figure 3A,B). The immunohistochemistry (IHC) detection of TNF-α revealed an increase in this cytokine in the MCT group compared with the control group, which was prevented with the allicin treatment (Figure 2C).

Second, to explore the mechanism through which allicin induces its anti-inflammatory effects, we determined the expressions of NFκB and Iκβ, its inhibitory protein. We found that MCT significantly upregulated NFκB compared with the control group (*p* < 0.05; Figure 4A), while the expression of this inflammatory protein was lower in the allicin-treated group compared with the MCT group (*p* < 0.05; Figure 4A). The expression of Iκβ in the MCT group tended to increase without statistically significant differences compared with the control group. However, in the group treated with allicin, there was a lower expression of Iκβ when compared with the MCT group (Figure 4B).

### 2.3. Antifibrotic Effects of Allicin

To assess the antifibrotic effects of allicin, we analyzed the fibrosis in the lungs using Masson’s trichrome staining. The results of this analysis demonstrated that fibrosis was significantly increased in the MCT group compared with the control group (Figure 5B vs. Figure 5A and Figure 5E vs. Figure 5D, respectively) (*p* < 0.05). As a consequence of allicin administration, fibrosis was prevented in the allicin-treated MCT group compared with the untreated MCT group (Figure 5C vs. Figure 5B and Figure 5F vs. Figure 5E, respectively) (*p* < 0.05).

To determine the possible antifibrotic mechanism of allicin on MCT-induced PAH, TGF-β and α-SMA were evaluated using Western blot analysis. First, we found that MCT significantly upregulated TGF-β and α-SMA compared with the control group (*p* < 0.05). However, with allicin treatment, the expression of these profibrotic markers was lower when compared with the MCT group (*p* < 0.05; Figure 6). In addition, we performed a second analysis using IHC to determine the specific site in which TGF-β was increased. IHC analysis revealed an increase in TGF-β in the pulmonary vessels of the MCT group compared with the control group, which was prevented by the allicin treatment (Figure 6C).

### 2.4. Effect of Allicin on Expression Levels of miR-21-5p and Signaling Pathway Bmpr2/Smad5

Furthermore, we determined the expression of miR-21-5p, an miRNA involved in the regulation of signaling pathways associated with the development of PAH. In this experiment, miR-21-5p was significantly upregulated in the MCT group compared with the control group (*p* < 0.05; Figure 7A). Allicin treatment did not modify the expression of miR-21-5p when compared with the MCT group (Figure 7A). On the other hand, we found a downregulation in the expression of Bmpr2 in the MCT and MCT + A groups compared with the control group (*p* < 0.05; Figure 7B). Smad5, as a transduction factor of the BMP signaling pathway, did not show significant differences between any of the analyzed groups (Figure 7C).

### 2.5. Effects of Allicin on Inflammation and Fibrosis in the RVs of MCT-Induced PAH Rats

RV hypertrophy is a predominant condition in PAH and is associated with damage due to the overexpression of proinflammatory cytokines and profibrotic proteins. Using this approach, we assessed the anti-inflammatory and antifibrotic effects of allicin on RV hyperthrophy in MCT-induced PAH by analyzing TNF-α, IL-6, and TGF-β. First, we determined the inflammatory markers. We found that TNF-α and IL-6 were increased in the MCT group compared with the control group (*p* < 0.05) (Figure 8A,B, respectively), while the increased expression of both cytokines was prevented by allicin treatment in the allicin-treated MCT group compared with the untreated MCT group (*p* < 0.05) (Figure 8A,B).

Finally, we determined the TGF-β levels in RV tissue from MCT-induced PAH rats. We found that MCT administration increased TGF-β levels compared with the control group (*p* < 0.05; Figure 9). Interestingly, the allicin treatment prevented the increase in TGF-β levels in the allicin-treated MCT group, resulting in the TGF-β levels being lower than those in the untreated MCT group (*p* < 0.05).

## 3. Discussion

In this study, we found that the oral administration of allicin induced a protective effect in MCT-induced PAH by preventing increased pulmonary arterial medial wall thickness and RV hypertrophy. In addition, we found that allicin treatment had anti-inflammatory and anti-fibrotic effects in PAH. In lung tissue, allicin induced low expressions of *Cd68*, TNF-α, IL-1β, IL-6, TGF-β, and α-SMA. In addition, allicin prevented increases in TNF-α, IL-6, and TGF-β in RV tissue, which is one of the heart’s four chambers and a determining organ related to the progression and severity of PAH.

Several mechanisms are involved in PAH progression, but the role of inflammation in triggering and maintaining pulmonary vascular remodeling has recently gained relevance [23]. However, therapies are aimed at stimulating vasodilation [24]. In the context of inflammation, several cytokines, such as TNF-α and IL-1β, are increased in PAH patients’ serum, which is related to low survival [25,26]. In addition, previous studies have reported that the expressions of TNF-α, IL-1β, and IL-6 are significantly increased in patients and experimental models of MCT-induced PAH [9,12]. In transgenic mice, the overexpression of TNF-α leads to the development of PAH while rats and dogs with MCT-induced PAH have elevated levels of TNF-α in the lung [27,28,29,30]. In addition, mice with an overexpression of IL-6 develop PAH, while knock-out mice for IL-6 do not develop the disease [31,32].

In the present study, the PAH model showed the characteristic damages of a previously reported model [33,34,35]. Thus, MCT induced RV hypertrophy and increased the arteriolar medial wall thickness. In this field, evidence suggests that pyrrolic derivatives of MCT that are metabolized in the liver induce pulmonary arterial endothelial cell (PAEC) damage through the activation of extracellular calcium-sensing receptors of PAECs, particularly its extracellular domain, which has the potential basic structure for MCT binding [36].

On the other hand, previous studies have demonstrated that allicin induces an anti-inflammatory effect [37,38]. It has been reported that allicin exerts an immune modulatory effect on intestinal epithelial cells through TNF-α inhibition [22]. In addition, allicin ameliorates the progression of osteoarthritis by decreasing TNF-α, IL-6, and IL-1β in chondrocytes [39]. In PAH, TNF-α, IL-6, and IL-1β lead to pulmonary arterial remodeling as they can cause damage in pulmonary endothelial cells, promoting abnormal PASMCs migration and proliferation [40,41]. Evidence also shows that the main feature of MCT-induced PAH is the infiltration of inflammatory cells and the secretion of inflammatory cytokines [42].

In this study, MCT administration increased the TNF-α, IL-6, and IL-1β levels in the lungs of rats with MCT-induced PAH. Furthermore, we found that MCT increased the expression of *Cd68*, an important macrophage marker [43]. Studies have shown that CD68^+^ levels are increased in experimental and clinical PAH, indicating cellular inflammation, which implies an increase in the number of perivascular macrophage infiltrations [44]. In addition, a recent study found that CD68^+^ macrophages are associated with the development of PAH [45]. Interestingly, allicin treatment prevented increases in the expressions of TNF-α, IL-6, IL-1β, and Cd68^+^ in the MCT model. These findings suggest that allicin reduces macrophage infiltration in the lung of rats with MCT-induced PAH and, consequently, ameliorates vascular remodeling. This could be supported by the immunohistochemistry analysis of TNF-α, which showed a low production of this cytokine in the MCT group treated with allicin.

To elucidate the possible anti-inflammatory mechanism of allicin on PAH, we studied the expression of NFκB, a transcription factor that has a key role in the expression of multiple genes associated with inflammation, proliferation, and apoptosis [46]. The activation of NFκB in cytoplasm is a consequence of Iκβ inhibitory protein phosphorylation and subsequent degradation by the proteasome. NFkB can migrate to the nucleus to induce the expression of cytokines (TNF-α, IL-6, and IL-1β), as well as proteins associated with cell proliferation and apoptosis, resulting in the development of PAH. Thus, to determine the mechanism through which allicin prevents increases in TNF-α, IL-6, IL-1β, and CD68^+^, we assessed the expression of NFκB in lung tissue. The result indicated that the protein expression levels of NFκB were lower in the allicin group than in the MCT group not treated with allicin. Other studies have demonstrated that, in MCT-induced PAH, the inhibition of NFκB improved the disease by decreasing macrophage infiltration [47]. Unexpectedly, we found that the Iκβ inhibitory protein was low in the MCT model with allicin treatment in comparison with the MCT model without allicin treatment. This could be possible because the phosphorylated form was not measured in this study. Thus, the results suggest that allicin may be considered a therapeutic alternative for inflammation in PAH through the modulation of proinflammatory cytokines and inhibition of inflammatory cell recruitment.

Remodeling of the pulmonary vasculature and fibrosis play key roles in the development and progression of PAH. In this context, TGF-β, α-SMA, fibronectin, and collagen are involved in the remodeling and fibrotic process [48,49]. Moreover, in PAH, the increase in TGF-β signaling results in the proliferation and antiapoptotic response of PAECs and PASMCs and in the increase in inflammatory cytokines [13]. Therefore, we assessed the expressions of TGF-β and α-SMA in MCT-induced PAH. Our results showed that MCT administration increased TGF-β and α-SMA in the lung tissue. Likewise, these data are in line with the analysis of fibrosis in lung tissue. Our results are in line with those previously reported [50]. Similar to inflammation, in PAH, there are no drugs that target fibrosis-related signaling pathways. Thus, we assessed the effects of allicin on fibrosis. A recent study reported that allicin decreased the TGF-β expression in the serum and renal cortex of rats with diabetic nephropathy [51]. In this work, we showed that allicin prevented increases in the expressions of TGF-β and α-SMA. Therefore, the results suggest that TGF-β and α-SMA contribute to fibrosis in the vascular wall. In addition to its role as a profibrotic protein, α-SMA is the first marker of differentiation of smooth muscle cells during remodeling of the vascular wall in PAH [52]. Thus, the upregulation of α-SMA could contribute to the muscularization of the vascular wall, the degree of vascular occlusion, and pulmonary artery medial wall thickness [53]. To the best of our knowledge, this is the first study to report the antifibrotic effect of allicin on PAH through the modulation of TGF-β and α-SMA. To support this result, we detected TGF-β using immunohistochemistry and found a lower production of this protein in the MCT group treated with allicin. Thus, the results suggest that allicin may be considered a therapeutic alternative for fibrosis in PAH.

On the other hand, miR-21-5p is an miRNA that plays a role in the development of PAH because it regulates the expressions of BMPR2 and TGF-β [54]. BMP signaling regulates cell proliferation, differentiation, and apoptosis [55], and it is decreased in patients with PAH, as well as in MCT models [56,57]. In this field, a decrease in BMP induces activation of the TGF-β signaling pathway [58]. Parikh et al. reported that miR-21-5p is upregulated in the lungs of rats with MCT-induced PAH [59]. In agreement with these results, we found that miR-21-5p was upregulated, while the expression of *Bmpr2* was downregulated. The allicin treatment did not modify the expressions of miR-21-5p and *Bmpr2*, and did not change *Smad5*, a transcription factor of BMP. These results suggest that allicin did not affect the *Bmpr2/smad5* signaling pathway via miR-21-5p in our experimental PAH model.

Besides the increases in inflammatory and fibrotic markers in the lung tissue, the PAH MCT model developed RV hypertrophy, which is also present in many PAH patients [34]. RV hypertrophy is a determining factor in the symptoms and survival of patients with PAH and is determined via the adaptation of RV function to the increased afterload [14]. Therefore, we assessed the expressions of inflammatory markers and fibrotic proteins in the RV. TNF-α, IL-6, and TGF-β were increased in the MCT group, and were prevented by the allicin treatment. Thus, the protective role of allicin appears to be significant and extended during the development of RV hypertrophy.

On the other hand, it is well known that the primary effects of allicin may be antioxidant and that the multiple cardioprotective effects attributed to the molecule could be due to an indirect effect. Allicin can react directly with reactive oxygen species (ROS) or free radicals or can act as a substrate for glutathione synthesis. This is supported by in vivo studies, which have reported that allicin reacts with glutathione to produce S-allylmercaptoglutathione or with L-cysteine to produce S-allylmercaptocysteine [60]. Moreover, allicin prevents the formation of free radicals and lipid peroxidation through hydroxyl and peroxyl radicals scavenging by transferring its allylic hydrogen to the oxidized substrate [61]. Indirectly, through regulation of the Nrf2/keap1 pathway and its target genes, allicin increases the presence of endogenous antioxidants, such as catalase, superoxide dismutase, heme-oxygenase, and glutathione peroxidase. At the same time, allicin regulates the secretion of proinflammatory cytokines by modulating NfκB/IκB pathway signaling [62,63,64,65]. Therefore, it is possible that the anti-inflammatory and antifibrotic effects observed in PAH could be associated with the antioxidant effects of allicin via modulation of the Nrf2/keap1 pathway. This issue could be addressed in another study.

Our study has some limitations, as follows. First, the allicin treatment started immediately after a single injection of MCT. Therefore, the effects of allicin on MCT-induced PAH could be preventive rather than curative. Second, we used an allicin dose that showed antidiabetic effects in other studies. Thus, it is possible that the effects of allicin in PAH could be dose-dependent. Third, another limitation of our study is the lack of a pulmonary hemodynamics parameter (RVSP, PVR, or PAP). However, the gold standard in MCT-induced PAH is the Fulton Index (RV/LV + S), which was assessed in our experimental model of PAH and was increased in the MCT group when compared with the control group. This index was in line with the histopathology analysis. Therefore, we conclude that the model was successfully induced. Our PAH validation results are in line with other reports in the literature [33,34,35]. Finally, this study is the first to explore the anti-inflammatory and antifibrotic effects of allicin (the major active component of garlic) in MCT-induced PAH.

Several studies have reported the beneficial effects of garlic in different presentations, such as extracts, lyophilization, and pills. Allicin has demonstrated a plethora of beneficial effects [66,67,68], but the dose used in experimental models, as well as in patients, is between 10 and 40 mg/day, and no secondary effects have been described. However, the use of allicin in patients is limited and focused on triglycerides and cholesterol alterations. Therefore, it is recommended to carry out controlled studies in patients in order to document scientific evidence to support the use of allicin in PAH.

In brief, this study showed evidence that allicin has a protective effect on pulmonary arterial medial wall thickness and RV hypertrophy in MCT-induced PAH. In addition, allicin prevented increases in inflammatory and fibrotic markers, which extended to the RV. Therefore, the nutraceutical allicin can be considered a potential therapeutic option, offering simultaneous and diverse benefits. Finally, further studies are required to show alternative mechanisms that help delay the progression of the disease.

## 4. Materials and Methods

### 4.1. Animals

Male Wistar rats weighing 200–250 g were used. The rats were randomly divided into three groups (*n* = 6 per group): control (CON), MCT-induced PAH (MCT), and MCT plus allicin (16/mg/kg/oral gavage technique) (MCT + A). The rats were kept under a chow diet (PMI Nutrition International, Brentwood, MO (www.labdiet.com/Products/StandardDiets), and water was given ad libitum, with 12 h light/dark cycles and a mean temperature of 22 °C. All experimental groups were maintained for 4 weeks. This protocol was approved by the Ethics Committee of the Instituto National de Cardiología Ignacio Chávez (Mexico) (INC/CICUAL/001/2021, approved on 27 January 2021) and handled in accordance with the regulations of the Mexican Official Norm (NOM-062-ZOO-1999) following the recommendations for the production, care, and use of laboratory animals.

### 4.2. Experimental Model of MCT-Induced PAH

The monocrotaline (MCT) (Catalogue C2401-1G Sigma) was dissolved in 1 N HCl and neutralized with 1 N NaOH. MCT administration was performed with a single subcutaneous injection of 60 mg/kg of body weight. All experimental groups were maintained for 4 weeks under similar conditions.

### 4.3. RV Hypertrophy

After 4 weeks, all rats were anesthetized by intraperitoneal injection with 50 mg/kg of ketamine and 10 mg/kg of xylazine. The complete lack of pain response was assessed by determining the pedal withdrawal reflex. Subsequently, a mid-thoracotomy was performed. The chest was opened, and after cardiac puncture to obtain blood, the heart and lung were cut. The tissues were placed in 0.9% saline solution to wash the blood. The heart was removed and separated from the RV and left plus ventricular septum (LV + S). The tissues were immediately frozen and kept at −70 °C for further analysis.

### 4.4. Histopathology

The pulmonary circulation was flushed with 5 mL of buffered saline at 37 °C. The left lung was prepared for morphometric analysis, and the right lung was removed and immediately frozen in liquid nitrogen and kept at −80 °C for further measurements. The left lung was flushed with saline to remove the blood and was fixed in 4% neutral buffered formalin for histology. The formalin-fixed lobes were subjected to paraffin embedding, were sectioned at 4 μm thickness with a microtome, were mounted on glass slides, were deparaffinized, and were stained with hematoxylin and eosin and with Masson’s trichrome according to common histopathological procedures. In each rat, 60 to 80 intra-acinar arteries with an external diameter of less than 50 μm were analyzed to calculate the medial wall thickness. The arterioles were photographed under 40× *g* magnification, and the morphometric analysis was performed with a histology automated system (Leica Microsystem Imaging Solutions LTD, Cambridge, UK). The percentage of wall thickness was calculated using the following formula: ((2 × medial wall thickness/external diameter) × 100)), and the muscularized vessel was determined as previously described [69]. Masson’s trichrome staining was used for pulmonary fibrosis. The percentage of pulmonary fibrosis (area ratio %) was calculated using the following formula: (blue stained area/total area of detection) × 100. Immunohistochemistry analysis of TNF-α and TGF-β was performed by taking the averages of the values obtained and by dividing the number of TNF-α and TGF-β positive cells by the total number of cells in each of the 5 different areas ((positive cells/total number of cells) × 100).

We determined the local production of TNF-α and TGF-β using immunohistochemistry. Heat-induced antigen retrieval was performed with ImmunoDNA Retrieve 1X with Citrate (Bio SB). Endogenous peroxidase was blocked with 3% hydrogen peroxide. The washes were carried out with 0.05% PBS-Tween 20. The tissue area was delimited and blocked with 200 μL of Background Sniper (Biocare Medical) and incubated for 20 min in a humid chamber. The tissues were incubated overnight at room temperature with shaking with (1) goat polyclonal antibody anti-TNF-α (1:100) (sc-1351, Santa Cruz Biotechnology) and (2) rabbit polyclonal antibody anti-TGF-β, (1:100) (sc-402, Santa Cruz Biotechnology). The tissues were washed, and rabbit PolyDetector HRP/DAB (BSB 0219, Bio SB) or Goat on Rodent HRP Polymer (GHP516L, Biocare Medical) was used as the secondary antibody and incubated for 30 min. In both cases, antibody binding was detected with 200 μL of diaminobenzidine (ImmPACT DAB Substrate Kit, Peroxidase) and counterstained with hematoxylin.

### 4.5. Analysis of Protein Expression

Lung (three randomly selected samples per group were analyzed) and heart tissue (four randomly selected samples per group were analyzed) were homogenized in lysis buffer (10 mM of HEPES, 0.2% Triton X-100, 50 mM of NaCl, 0.5 mM of sucrose, 0.1 mM of EDTA, protease, and phosphatase inhibitors). The homogenate was centrifuged at 10,000 rpm for 10 min at 4 °C. The supernatants were separated, aliquoted, and stored at −70 °C. Total protein concentration in the samples was determined by the Bradford method using bovine serum albumin as the standard. The proteins (7.5 µg) were resolved by SDS-PAGE and electrotransferred onto a polyvinylidene fluoride membrane (Millipore Corp., Bedford, MA). The following antibodies from Santa Cruz Biotechnology (Santa Cruz Biotechnology, Inc., Dallas, TX, USA) were used: interleukin (IL)-1β (IL-1β): sc-1250 (1:5000), IL-6: sc-57315 (1:12,000), TNF-α: sc-52746 (1:12,000), and TGF-β: sc-130348 (1:10,000). The following antibodies from GTX (Gene Tex) were used: NFκB p65: 102090 (1:2000) and Iκβ: 82797 (1:5000). The alpha-smooth muscle actin (α-SMA): ab265588 (1:5000) antibody from ABCAM (Cambridge, MA, USA) was used. The detection of the primary antibody was carried out with a horseradish peroxidase-conjugated secondary antibody and enhanced chemiluminescence reagents (Clarity Western ECL Substrate, Bio-Rad). Positive immunoreactive bands were quantified using a Kodak Electrophoresis Documentation and Analysis System 290 (EDAS 290). For loading controls, Coomassie Blue R-250 (Bio-Rad, Hercules, CA, USA) staining was used. The protein expression was expressed as the ratio of the protein of interest in the sample to that in the loading control in arbitrary units (a.u.).

### 4.6. miRNAs and mRNA Determination by RT-qPCR

Total RNA was extracted from 80 mg of lung tissue isolated with the Tripure reagent according to the manufacturer’s recommendations (Roche, Basel, Switzerland). The total RNA obtained was quantified and later used to evaluate the expressions of the mRNA of *Bmpr2*, *Smad5*, and *Cd68*, as well as the expressions of miR-21-5p and U87. The isolated RNA was immediately converted to cDNA, as described below.

MicroRNAs were determined using two-step RT-qPCR with an RT-primer specific assay in combination with TaqMan probes, namely miR-21-5p (ID:000397) and U87 (ID:001712) (Applied Biosystems, CA, USA). Each RT-reaction used 1.5 µL from the 14 µL eluted RNA using the TaqMan MicroRNA Reverse Transcription Kit (Applied Biosystems, CA, USA). In addition, the expression levels of the mRNA of *Bmpr2*, *Smad5*, and *Cd68* were determined using two-step RT-qPCR with random hexamers in combination with TaqMan probes from the Universal Probe Library (Roche; Basel, Switzerland) (see Table 1). In total, 500 ng of extracted RNA was used for cDNA synthesis using the QuantiNova Reverse Transcription Kit (Qiagen, Hilden, Germany). The RT reaction program and PCR cycling conditions were as previously reported [70]. PCR was performed using a LightCycler TM 480 II System (Roche; Basel, Switzerland) with the QuantiNova Probe PCR Kit (Qiagen, Hilden, Germany). In addition, miR-21-5p relative concentrations were normalized with Ct values of U87, and mRNA relative concentrations were normalized with Ct values of *Rplp0* for tissue (Applied Biosystems, CA, USA). The values were calculated using the 2^−ΔΔCt^ method.

### 4.7. Statistical Analysis

The data are presented as means and standard errors. Accordingly, the differences between groups were assessed by ordinary one-way ANOVA followed by the Tukey multiple comparison test (*p* < 0.05) using the Graph Pad Prism software version 6.

## 5. Conclusions

The results showed that allicin prevented the increase in pulmonary arterial medial wall thickness and RV hypertrophy in MCT-induced PAH, which was possibly mediated by its effects on inflammatory and fibrosis markers in the lung and heart tissues. Therefore, allicin is a nutraceutical offering diverse benefits and should be considered as a potential therapeutic option to delay pulmonary function decline and right ventricle hypertrophy in the progression of PAH.

## Figures and Tables

**Figure 1 ijms-22-08600-f001:**
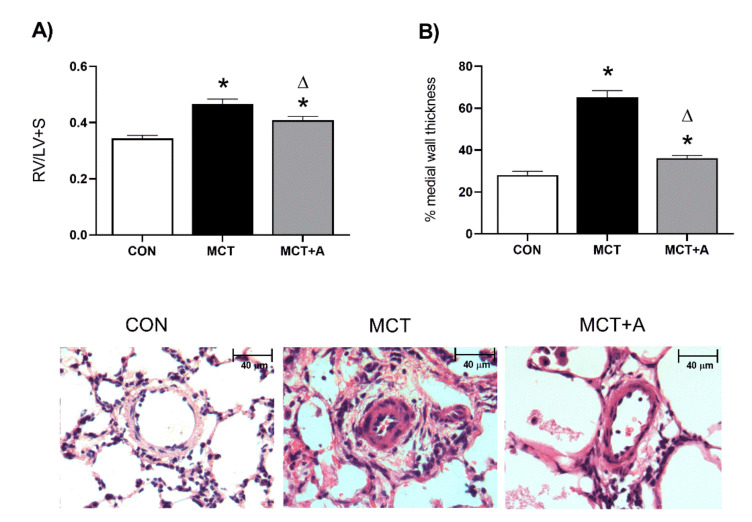
Pulmonary arterial hypertension markers. (**A**) Effect of allicin on RV hypertrophy in MCT-induced PAH rats. RV hypertrophy was measured using the Fulton index RV/LV + S (%). Control group (CON), saline solution as the vehicle; monocrotaline group (MCT) with 60 mg/kg; and MCT with 16 mg/kg of allicin (MCT + A). (**B**) Effect of allicin on pulmonary arteries wall in MCT-induced PAH rats. Differences were tested by ordinary one-way ANOVA followed by the Tukey multiple comparison test. Data are presented as mean and standard errors. *n* = 6, * *p* < 0.05 vs. control; ∆ *p* < 0.05 vs. MCT.

**Figure 2 ijms-22-08600-f002:**
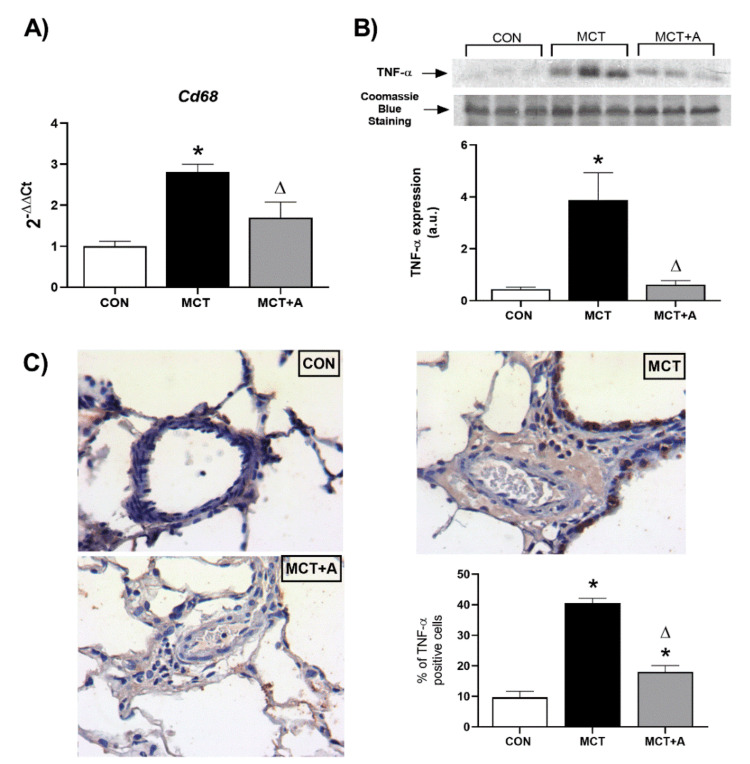
Effect of allicin on inflammatory mediators in MCT-induced PAH rats. (**A**) mRNA expression of Cd68 and (**B**) protein expression of TNF-α. For Western blotting, three randomly selected samples per group were analyzed. (**C**) Immunohistochemistry analysis of TNF-α. Control group (CON), saline solution as the vehicle; monocrotaline group (MCT) with 60 mg/kg; and MCT with 16 mg/kg of allicin (MCT + A). Differences were tested by ordinary one-way ANOVA followed by the Tukey multiple comparison test. Data are presented as mean and standard errors. *n* = 6, * *p* < 0.05 vs. Control; ∆ *p* < 0.05 vs. MCT.

**Figure 3 ijms-22-08600-f003:**
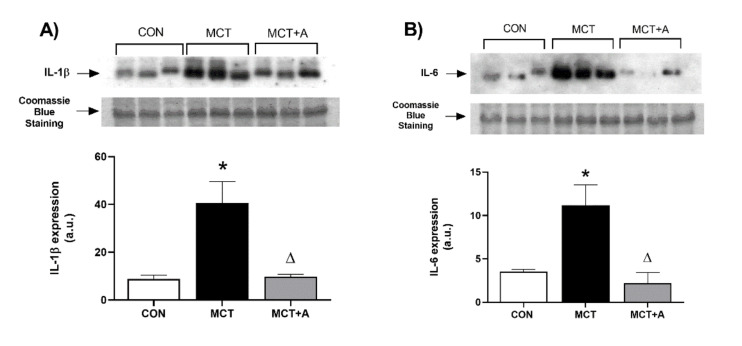
Effect of allicin on inflammatory cytokines in MCT-induced PAH rats. (**A**) IL-1β protein expression and (**B**) IL-6 protein expression. Control group (CON), saline solution as the vehicle; monocrotaline group (MCT) with 60 mg/kg; and MCT with 16 mg/kg of allicin (MCT + A). For Western blotting, three randomly selected samples per group were analyzed. Differences were tested by ordinary one-way ANOVA followed by the Tukey multiple comparison test. Data are presented as mean and standard errors. * *p* < 0.05 vs. Control; ∆ *p* < 0.05 vs. MCT.

**Figure 4 ijms-22-08600-f004:**
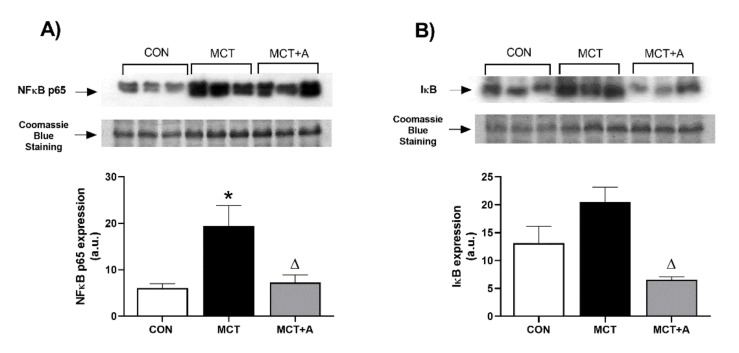
Effect of allicin on NFκB p65 and Iκβ in MCT-induced PAH rats. (**A**) NFκB p65 protein expression and (**B**) Iκβ protein expression. Control group (CON), saline solution as the vehicle; monocrotaline group (MCT) with 60 mg/kg; and MCT with 16 mg/kg of allicin (MCT + A). For Western blotting, three randomly selected samples per group were analyzed. Differences were tested by ordinary one-way ANOVA followed by the Tukey multiple comparison test. Data are presented as mean and standard errors. * *p* < 0.05 vs. Control; ∆ *p* < 0.05 vs. MCT.

**Figure 5 ijms-22-08600-f005:**
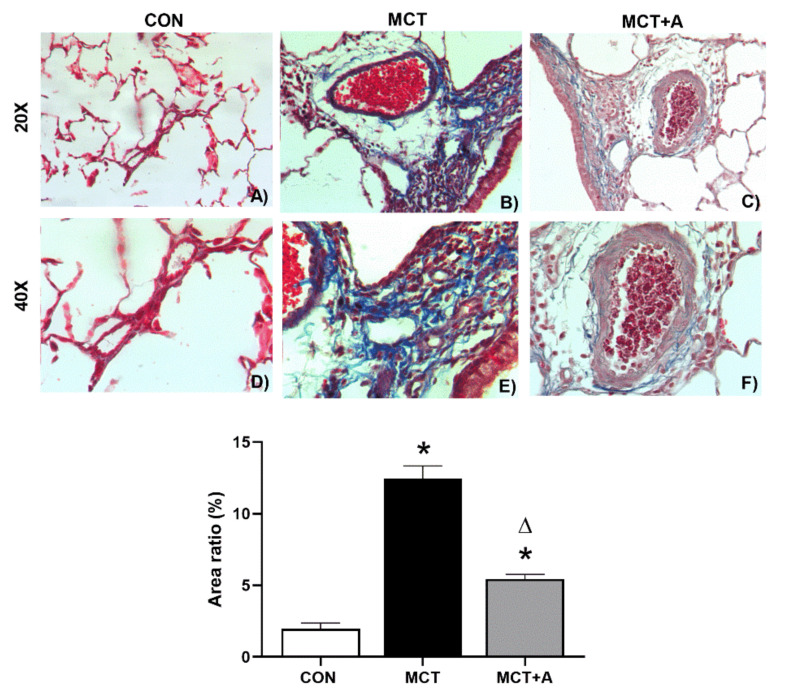
Lung fibrosis in MCT-induced PAH rats. Control group (CON), saline solution as the vehicle; monocrotaline group (MCT) with 60 mg/kg; and MCT with 16 mg/kg of allicin (MCT + A). (**A**–**C**) Original magnification 20X. (**D**–**F**) Original magnification 40X. Differences were tested by ordinary one-way ANOVA followed by the Tukey multiple comparison test. Data are presented as mean and standard errors. *n* = 6, * *p* < 0.05 vs. Control; ∆ *p* < 0.05 vs. MCT.

**Figure 6 ijms-22-08600-f006:**
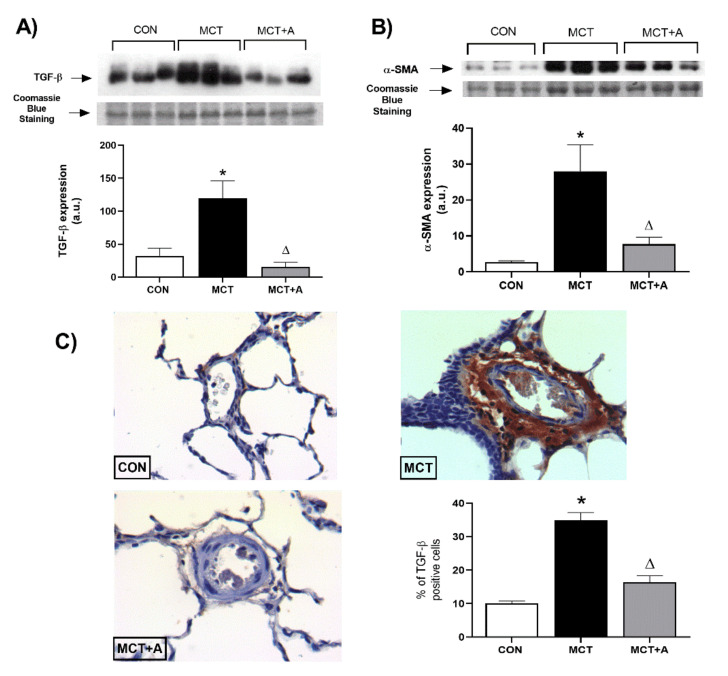
Effect of allicin on the protein expression of TGF-β and α-SMA in MCT-induced PAH rats. (**A**) TGF-β protein expression and (**B**) α-SMA protein expression. For Western blotting, three randomly selected samples per group were analyzed. (**C**) Immunohistochemical analysis of TGF-β. Control group (CON), saline solution as the vehicle; monocrotaline group (MCT) with 60 mg/kg; and MCT with 16 mg/kg of allicin (MCT + A). Differences were tested by ordinary one-way ANOVA followed by the Tukey multiple comparison test. Data are presented as mean and standard errors. *n* = 6, * *p* < 0.05 vs. Control; ∆ *p* < 0.05 vs. MCT.

**Figure 7 ijms-22-08600-f007:**
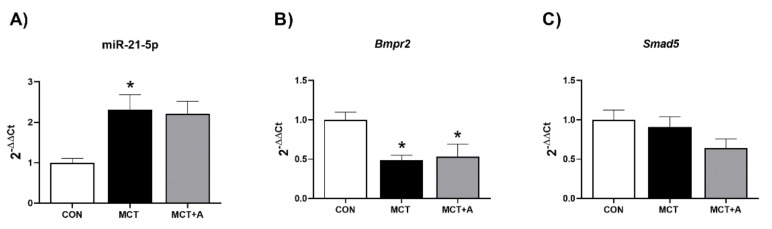
Effect of allicin on miR-21-5p and signal pathway Bmpr2/Smad5 in MCT-induced PAH rats. (**A**) miR-21-5p expression, (**B**) mRNA expression of Bmpr2, and (**C**) mRNA expression of Smad5. Control group (CON), saline solution as the vehicle; monocrotaline group (MCT) with 60 mg/kg; and MCT with 16 mg/kg of allicin (MCT + A). Differences were tested by ordinary one-way ANOVA followed by Tukey’s multiple comparison test. Data are presented as mean and standard errors. *n* = 6, * *p* < 0.05 vs. Control; ∆ *p* < 0.05 vs. MCT.

**Figure 8 ijms-22-08600-f008:**
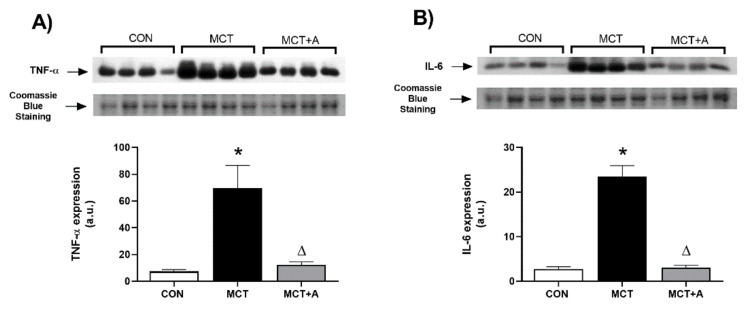
Effect of allicin on inflammatory mediators in the RVs of MCT-induced PAH rats. (**A**) TNF-α protein expression and (**B**) IL-6 protein expression. Control group (CON), saline solution as the vehicle; monocrotaline group (MCT) with 60 mg/kg; and MCT with 16 mg/kg of allicin (MCT + A). For Western blotting, four randomly selected samples were analyzed per group. Differences were tested by ordinary one-way ANOVA followed by the Tukey multiple comparison test. Data are presented as mean and standard errors. * *p* < 0.05 vs. Control; ∆ *p* < 0.05 vs. MCT.

**Figure 9 ijms-22-08600-f009:**
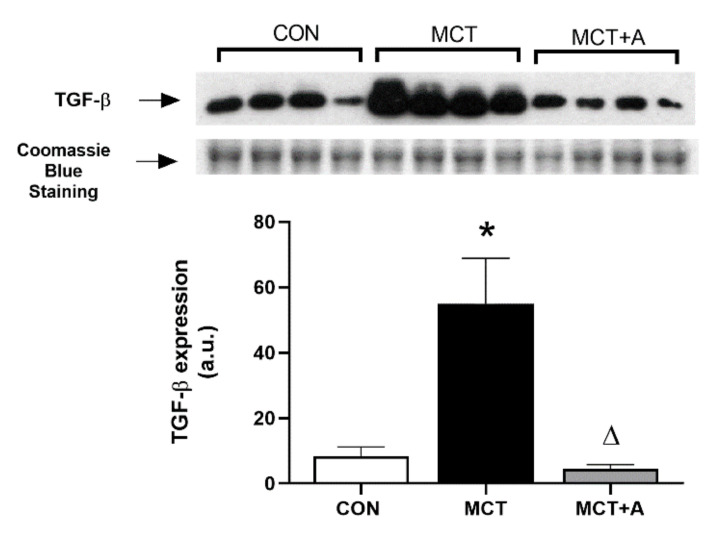
Effect of allicin on TGF-β levels in the RVs of MCT-induced PAH rats. Control group (CON), saline solution as the vehicle; monocrotaline group (MCT) with 60 mg/kg; and MCT with 16 mg/kg of allicin (MCT + A). For Western blotting, four randomly selected samples per group were analyzed. Differences were tested by ordinary one-way ANOVA followed by the Tukey multiple comparison test. Data are presented as mean and standard errors. * *p* < 0.05 vs. Control; ∆ *p* < 0.05 vs. MCT.

**Table 1 ijms-22-08600-t001:** Primer sequences used for RT-qPCR analysis.

Gene	GenBank ID	Direction	Primer (5′-3′)	UPL
*Bmpr2*	NM_080407.1	Forward	gagccctccctggacttg	67
		Reverse	atatcgaccccgtccaatc	
*Smad5*	NM_021692.1	Forward	gcctatggacacaagcaaca	107
		Reverse	aggcaacaggctgaacatct	
*Cd68*	NM_001031638.1	Forward	cgccagtgaccaatctctc	34
		Reverse	gggtaacgcagaaggcaat	
*Rplp0*	NM_022402.2	Forward	gatgcccagggaagacag	85
		Reverse	gaagcattttgggtagtcatcc	

## Data Availability

The data presented in this study are available on request from the corresponding author.

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
