# Peer review of "Anti-Inflammatory Effect of Allicin Associated with Fibrosis in Pulmonary Arterial Hypertension"

_ijms, 2021, doi:10.3390/ijms22168600_

Round 1

Reviewer 1 Report

This study assesses the role of allicin on inflammatory markers in a monocrotaline model of pulmonary hypertension, examining responses in both the lung and heart. Although numerous effects of allicin are observed, little mechanistic insight into these observations are provided. Furthermore, the study is lacking in controls examining the effects of allicin in normotensive rats. Complete comments are detailed below.

Broad Comments:

  • The authors show the effects of allicin on a wide variety of markers of inflammatory markers in the lung and right ventricle of monocrotaline-treated rats. However, the mechanism by which allicin reduces these indices is unaddressed.
  • There are numerous errors in English language that limit the clarity of the manuscript.

Specific Comments:

Major concerns:

  • Within the methods the authors state that a n=6 was used for each group, however the specific number of observations for each figure is not clear. This could be clarified within the figure legends.
  • The authors are missing the key control examining the effects of allicin in lungs and right ventricles of control rats. This experimental group is critical to examine whether the alterations due to allicin are only present in the disease model.
  • The resolution of the representative blots seems very low. This makes bands harder to discern, particularly in figures such as Figure 2.
  • The authors state in section 2.2 that immunohistochemistry reveals and increase in TNF-α in MCT rats which is decreased by allicin. Showing quantification of data (with statistics) is necessary to substantiate this point.
  • Some discussion of the limitation of whether increased α-SMA in MCT is only a fibrotic marker or could also be influenced by arterial wall thickening (Figure 1B) should be included. It would also be beneficial to quantify immunohistochemical staining of TGF-β.
  • The authors mention the antioxidant properties of allicin, however do not investigate whether the effects of the allicin are mediated by these properties alone, or through other mechanisms.
  • There are numerous errors in English language manuscript. It may benefit the authors to seek professional editing services.

Minor concerns:

  • Abstract: Seems as though the word “characterized” is missing between “is” and “by” in the 1st
  • Results 2.1, line 4: Not sure the what is meant by the word “capered”.
  • Methods 4.2, line 1: Should read “1 M HCl”.
  • Table 1 in methods: remove extra numbers from right hand column.
  • Suggest moving the (5) conclusions to after the discussion rather than after the methods.

Author Response

REVIEWER 1

Comments and Suggestions for Authors

This study assesses the role of allicin on inflammatory markers in a monocrotaline model of pulmonary hypertension, examining responses in both the lung and heart. Although numerous effects of allicin are observed, little mechanistic insight into these observations are provided. Furthermore, the study is lacking in controls examining the effects of allicin in normotensive rats. Complete comments are detailed below.

Answer: We thank the reviewer for their valuable time and comments. We are including a point-by-point response to the comments. Additionally, we have included the possible mechanisms involved for the allicin protection, into the discussion section. 

Broad Comments:

1.- The authors show the effects of allicin on a wide variety of markers of inflammatory markers in the lung and right ventricle of monocrotaline-treated rats. However, the mechanism by which allicin reduces these indices is unaddressed.

Answer: The reviewer's comment is spot on. Thus, to address this comment, we have rewritten the discussion section on page 10, lines 303 to 320

“To elucidate the possible anti-inflammatory mechanism of allicin on PAH, we studied the expression of NFкB, a transcription factor that has a key role in the expression of multiple genes associated with inflammation, proliferation, and apoptosis [46]. The activation of NFкB in cytoplasm is a consequence of Iкb inhibitory protein phosphorylation and subsequent degradation by the proteasome. NFkB can migrate to the nucleus to induce the expression of cytokines (TNF-α, IL-6, and IL-1β), as well as proteins associated with cell proliferation and apoptosis, resulting in the development of PAH. Thus, to determine the mechanism through which allicin prevents increases in TNF-a, IL-6, IL-1b, and CD68+, we assessed the expression of NFкB in lung tissue. The result indicated that the protein expression levels of NFкB were lower in the allicin group than in the MCT group not treated with allicin. Other studies have demonstrated that, in MCT-induced PAH, the inhibition of NFкB improved the disease by decreasing macrophage infiltration [47]. Unexpectedly, we found that the Iкb inhibitory protein was low in the MCT model with allicin treatment in comparison with the MCT model without allicin treatment. This could be possible because the phosphorylated form was not measured in this study. Thus, the results suggest that allicin may be considered a therapeutic alternative for inflammation in PAH through the modulation of proinflammatory cytokines and inhibition of inflammatory cell recruitment.”

2.- There are numerous errors in English language that limit the clarity of the manuscript.

 Answer: As suggested by the reviewer, we submitted the manuscript for grammar edition. 

Specific Comments:

Major concerns:

1.- Within the methods the authors state that a n=6 was used for each group, however the specific number of observations for each figure is not clear. This could be clarified within the figure legends.

Answer: We thank the reviewer for her/his careful review and comments.

In agreement with the reviewer's comment, in this reviewed version of our manuscript, we have included the number of rats in each experimental group in the methods section as well as in the respective legend figure caption.

2.- The authors are missing the key control examining the effects of allicin in lungs and right ventricles of control rats. This experimental group is critical to examine whether the alterations due to allicin are only present in the disease model.

Answer: Thank you for your comment.

The main objective of the study was to evaluate the effects of allicin on PAH development. However, other studies reported that allicin administered to a control group showed no changes in systolic blood pressure, heart rate, gross and whole hearts, neither changes in hypertrophic markers compared with the control untreated 1. Furthermore, in previous trials to determine the toxicity tests on healthy organisms (doses greater than those used), we used allicin at lower (8 mg/kg) and higher doses (32 mg/kg) than used in our study. We did not observe mortality (data not included), which suggests low toxicity of allicin at the doses as used in our study (16 mg/kg).

Thus, based on our experimental evidence as well as the other studies, we did not include a control group treated with allicin. On the other hand, we observed decreased expression of proinflammatory and profibrotic markers in lung and RV in the allicin treated group, which suggests a predominant beneficial effect of allicin instead of a side effect.  

1.- Liu C, Cao F, Tang QZ, Yan L, Dong YG, Zhu LH, Wang L, Bian ZY, Li H. Allicin protects against cardiac hypertrophy and fibrosis via attenuating reactive oxygen species-dependent signaling pathways. J Nutr Biochem. 2010 Dec;21(12):1238-50. doi: 10.1016/j.jnutbio.2009.11.001. Epub 2010 Feb 25. PMID: 20185286.

3.- The resolution of the representative blots seems very low. This makes bands harder to discern, particularly in figures such as Figure 2.

Answer: We thank the reviewer for his/her valuable observation. In the present version of the manuscript we have analyzed again the results and improved the resolution of the blot.

4.- The authors state in section 2.2 that immunohistochemistry reveals and increase in TNF-α in MCT rats which is decreased by allicin. Showing quantification of data (with statistics) is necessary to substantiate this point.

Answer: We thank the reviewer for his/her valuable observation. In the present version of the manuscript we have included the graph of densitometry analysis of immunohistochemistry from TNFa.

5.- Some discussion of the limitation of whether increased α-SMA in MCT is only a fibrotic marker or could also be influenced by arterial wall thickening (Figure 1B) should be included. It would also be beneficial to quantify immunohistochemical staining of TGF-β.

Answer: We thank the reviewer for her/his careful review and comments. We have included the graph of densitometry analysis of immunohistochemistry from TGF-b in the results section and a brief discussion about the role of a-SMA in the vascular remodeling wall in the discussion section.

“Therefore, the results suggest that TGF-b and α-SMA contribute to fibrosis in the vascular wall. In addition to its role as a profibrotic protein, α-SMA is a first marker of differentiation of smooth muscle cells during remodeling of the vascular wall in PAH [52]. Thus, the upregulation of α-SMA could contribute to the muscularization of the vascular wall, the degree of vascular occlusion, and pulmonary artery medial wall thickness [53]. ”

6.- The authors mention the antioxidant properties of allicin, however do not investigate whether the effects of the allicin are mediated by these properties alone, or through other mechanisms.

Answer: We thank the reviewer for her/his careful review and comments. To address this comment, we have included a paragraph in the discussion section on page 11, lines 415 to 429

“On the other hand, it is well known that the primary effects of allicin may be antioxidant and that the multiple cardioprotective effects attributed to the molecule could be due to an indirect effect. Allicin can react directly with reactive oxygen species (ROS) or free radicals or can act as a substrate for glutathione synthesis. This is supported by in vivo studies, which have reported that allicin reacts with glutathione to produce S-Allyl-mercaptoglutathione or with L-cysteine to produce S-allyl-mercaptocysteine [60]. Moreover, allicin prevents the formation of free radicals and lipid peroxidation through hydroxyl and peroxyl radicals scavenging by transferring its allylic hydrogen to the oxidized substrate [61]. Indirectly, through regulation of the Nrf2/keap1 pathway and its target genes, allicin increases the presence of endogenous antioxidants, such as catalase, superoxide dismutase, heme-oxygenase, and glutathione peroxidase. At the same time, allicin regulates proinflammatory cytokines secretion by modulating NfkB/Ikb pathway signaling [62–65]. Therefore, it is possible that the anti-inflammatory and antifibrotic effects observed in PAH could be associated with the antioxidant effects of allicin via modulation of the Nrf2/keap1 pathway. This issue could be addressed in another study.”

7.-There are numerous errors in English language manuscript. It may benefit the authors to seek professional editing services.

 Answer: As suggested by the reviewer, we submitted the manuscript for grammar edition. 

Minor concerns:

1.- Abstract: Seems as though the word “characterized” is missing between “is” and “by” in the 1st

Answer: We thank the reviewer's comments. We have changed to the correct phrase

2.- Results 2.1, line 4: Not sure the what is meant by the word “capered”.

Answer: We thank the reviewer for her/his careful review and comments. The change has been made

3.- Methods 4.2, line 1: Should read “1 M HCl”.

Answer: We regret the confusion with the concentration units, according with Chaumais et al, MCT could be dissolve in 1 N HCl.

However, for all solutions which have gram equivalent value of 1 (ions dissociate), the normality of the solution is always equal to the molarity of the solution. Thus, in HCl Normality (N) = Molarity (M).

Chaumais MC, Ranchoux B, Montani D, Dorfmüller P, Tu L, Lecerf F, Raymond N, Guignabert C, Price L, Simonneau G, Cohen-Kaminsky S, Humbert M, Perros F. N-acetylcysteine improves established monocrotaline-induced pulmonary hypertension in rats. Respir Res. 2014 Jun 14;15(1):65. doi: 10.1186/1465-9921-15-65. PMID: 24929652; PMCID: PMC4065537.

4.- Table 1 in methods: remove extra numbers from right hand column.

Answer: We thank the reviewer for her/his careful review and comments. The change has been made

5.- Suggest moving the (5) conclusions to after the discussion rather than after the methods.

Answer: We thank the reviewer for her/his careful review and comments. The change has been made

Reviewer 2 Report

The manuscript seems to cover an interesting topic as it evaluates the potential anti-inflammatory effect of plant-derived allicin on the pulmonary arterial medial wall thickness and RV hypertrophy in MCT-induced PAH with special attention to the mechanistic background of such beneficial activity. The presented data are wide, and methodology appears appreciate. Some comments/questions can include as follows:

  • The authors could specify (Methods, discussion) if it was preventive or therapeutic regimen. According to Methods it seems that allicin administration was started immediately after single injection of MCT, please provide rationale for preventive animal PH model that was chosen for such observations rather than therapeutic one, where the potential drug efficacy concerns reversal – not only prevention of PH-linked lesions.
  • Methods: If any procedures were blinded?
  • Methods: If allicin was administered by oral gavage? Please complete
  • Methods: Please state whether the data were distributed normally and if the homogeneity of variance was checked for ANOVA
  • Results: If such beneficial effect of allicin concerned also improvement in pulmonary heamodynamics? – please refer to missing data on RVSP, PVR or PAP
  • Results: Please complete the description of figures according to strict data about number of animals (per group) that were included into the particular analysis – the captions of Fig 1-9. The survival analysis would be valuable addition to current data
  • What are clinical implications of the current study e.g., corresponding dose used in animals and humans. A majority of preclinical studies indicate potential efficacy of plant-derived agents in PH without its further confirmation during clinical trials. The authors could strengthen rationale and novelty of the current survey and obtained results.
  • What are the limitations of the study

Author Response

Reviewer 2

Comments and Suggestions for Authors

The manuscript seems to cover an interesting topic as it evaluates the potential anti-inflammatory effect of plant-derived allicin on the pulmonary arterial medial wall thickness and RV hypertrophy in MCT-induced PAH with special attention to the mechanistic background of such beneficial activity. The presented data are wide, and methodology appears appreciate. Some comments/questions can include as follows:

1.- The authors could specify (Methods, discussion) if it was preventive or therapeutic regimen. According to Methods it seems that allicin administration was started immediately after single injection of MCT, please provide rationale for preventive animal PH model that was chosen for such observations rather than therapeutic one, where the potential drug efficacy concerns reversal – not only prevention of PH-linked lesions.

Answer: We thank the reviewer for her/his careful review and comments.

To addres this comment, in the  discussion section (page 11, lines 430to 432)  we have included  a brief description.

“Our study has some limitations, as follows. First, the allicin treatment started immediately after a single injection of MCT. Therefore, the effects of allicin on MCT-induced PAH could be preventive rather than curative”.

Also, we have changed the description and discussion of our results “prevented” rather than “decreased”, to emphasize a preventive effect, not curative

On the other hand, this administration scheme was implemented based on studies assessing the effect of other potential therapeutic candidates for PAH treatment 1, 2, 3.   The results showed beneficial effects induced by the allicin treatment which would be interpreted as preventive, not as a curative. Therefore our results and conclusion were described as such. However, the observation of the reviewer is right and could be addressed in a future work.

 1.-Liu C, Cao F, Tang QZ, Yan L, Dong YG, Zhu LH, Wang L, Bian ZY, Li H. Allicin protects against cardiac hypertrophy and fibrosis via attenuating reactive oxygen species-dependent signaling pathways. J Nutr Biochem. 2010 Dec;21(12):1238-50. doi: 10.1016/j.jnutbio.2009.11.001. Epub 2010 Feb 25. PMID: 20185286.

2.- Bombicz, M., Priksz, D., Varga, B., Kurucz, A., Kertész, A., Takacs, A., Posa, A., Kiss, R., Szilvassy, Z., & Juhasz, B. (2017). A Novel Therapeutic Approach in the Treatment of Pulmonary Arterial Hypertension: Allium ursinum Liophylisate Alleviates Symptoms Comparably to Sildenafil. International journal of molecular sciences18(7), 1436. https://doi.org/10.3390/ijms18071436

3.- Park, B. M., Chun, H., Chae, S. W., & Kim, S. H. (2017). Fermented garlic extract ameliorates monocrotaline-induced pulmonary hypertension in rats. Journal of Functional Foods30, 247-253. https://doi.org/10.1016/j.jff.2017.01.024

2.- Methods: If any procedures were blinded?

Answer: The mRNA expression, histopathology, as well as the immunohistochemistry analysis, were made blind

3.-Methods: If allicin was administered by oral gavage? Please complete

Answer: In agreement with the reviewer comment, now we have included the administration route in the methods section (page 12, line 463)

“allicin (16 mg/kg/oral gavage technique)”

4.-Methods: Please state whether the data were distributed normally and if the homogeneity of variance was checked for ANOVA

Answer: With exception of the protein expression analysis, all the data meet with the normality and homogeneity test checked by ANOVA available homogeneity variance test. 

Data are presented as means and standard errors. Accordingly, differences between groups were assessed by ordinary one-way ANOVA followed by Tukey multiple comparison test (p <0.05) using the Graph Pad Prism software version 6.

5.- Results: If such beneficial effect of allicin concerned also improvement in pulmonary heamodynamics? – please refer to missing data on RVSP, PVR or PAP

Answer: We thank the reviewer for her/his careful review and comments. To addres this comment, in the  discussion section (page 11, lines 430 to 440)  we have included  a brief description.

“Our study has some limitations, as follows. First, the allicin treatment started immediately after a single injection of MCT. Therefore, the effects of allicin on MCT-induced PAH could be preventive rather than curative. Second, we used an allicin dose that showed antidiabetic effects in other studies. Thus, it is possible that the effects of allicin in PAH could be dose-dependent. Third, another limitation of our study is the lack of a pulmonary hemodynamics parameter (RVSP, PVR, or PAP). However, the gold standard in MCT-induced PAH is the Fulton Index (RV/LV+S), which was assessed in our experimental model of PAH and was increased in the MCT group when compared with the control group. This index was in line with the histopathology analysis. Therefore, we conclude that the model was successfully induced. Our PAH validation results are in line with other reports in the literature [33–35].”

6.- Results: Please complete the description of figures according to strict data about number of animals (per group) that were included into the particular analysis – the captions of Fig 1-9. The survival analysis would be valuable addition to current data

Answer: We thank the reviewer for her/his careful review and comments.

In agreement with the reviewer's comment, in this reviewed version of our manuscript, we have included the number of rats in each experimental group in the methods section, as well as in the respective legend figure caption.

7.- What are clinical implications of the current study e.g., corresponding dose used in animals and humans. A majority of preclinical studies indicate potential efficacy of plant-derived agents in PH without its further confirmation during clinical trials. The authors could strengthen rationale and novelty of the current survey and obtained results.

Answer: We thank the reviewer for her/his careful review and comments. Based on scientific literature supporting the use of allicin we have included a paragraph with the Clinical implications into the discussion section (page 12, line 446 to 452).

“Several studies have reported the beneficial effects of garlic in different presentations, such as extracts, lyophilization, and pills. Allicin has demonstrated a plethora of beneficial effects [66–68], but the dose used in experimental models as well as in patients is between 10 and 40 mg/day, and no secondary effects have been described. However, the use of allicin in patients is limited and focused on triglycerides and cholesterol alterations. Therefore, it is recommended to carry out controlled studies in patients in order to document scientific evidence to support the use of allicin in PAH.”.

8.- What are the limitations of the study

Answer: We thank the reviewer for her/his careful review and comments. We have included the limitations into the discussion section (page 11, line 430 to 440).

“Our study has some limitations, as follows. First, the allicin treatment started immediately after a single injection of MCT. Therefore, the effects of allicin on MCT-induced PAH could be preventive rather than curative. Second, we used an allicin dose that showed antidiabetic effects in other studies. Thus, it is possible that the effects of allicin in PAH could be dose-dependent. Third, another limitation of our study is the lack of a pulmonary hemodynamics parameter (RVSP, PVR, or PAP). However, the gold standard in MCT-induced PAH is the Fulton Index (RV/LV+S), which was assessed in our experimental model of PAH and was increased in the MCT group when compared with the control group. This index was in line with the histopathology analysis. Therefore, we conclude that the model was successfully induced. Our PAH validation results are in line with other reports in the literature [33–35].”   

Round 2

Reviewer 1 Report

This study assesses the role of allicin on inflammatory markers in a monocrotaline model of pulmonary hypertension, examining responses in both the lung and heart. The presentation and discussion of findings has been greatly improved in the revised manuscript.